# Semi-Supervised Medical Image Segmentation via Cross Teaching between CNN and Transformer

**Xiangde Luo**[1,2]                                    XIANGDE.LUO@STD.UESTC.EDU.CN
**Minhao Hu**[3]                                          HUMINHAO@SENSETIME.COM
**Tao Song**[3]                                              SONGTAO@SENSETIME.COM
**Guotai Wang**[1,2,*]                                        GUOTAI.WANG@UESTC.EDU.CN
**Shaoting Zhang**[1,2,3]                             ZHANGSHAOTING@UESTC.EDU.CN

[1] *University of Electronic Science and Technology of China, Chengdu, China*

[2] *Shanghai AI Lab, Shanghai, China;*

[3] *SenseTime, Shanghai, China;*

[*] *Corresponding author*

## Abstract

Recently, deep learning with Convolutional Neural Networks (CNNs) and Transformers has shown encouraging results in fully supervised medical image segmentation. However, it is still challenging for them to achieve good performance with limited annotations for training. This work presents a very simple yet efficient framework for semi-supervised medical image segmentation by introducing the cross teaching between CNN and Transformer. Specifically, we simplify the classical deep co-training from consistency regularization to cross teaching, where the prediction of a network is used as the pseudo label to supervise the other network directly end-to-end. Considering the difference in learning paradigm between CNN and Transformer, we introduce the Cross Teaching between CNN and Transformer rather than just using CNNs. Experiments on a public benchmark show that our method outperforms eight existing semi-supervised learning methods just with a more straightforward framework. Notably, this work may be the first attempt to combine CNN and transformer for semi-supervised medical image segmentation and achieve promising results on a public benchmark. Code is available at: https://github.com/HiLab-git/SSL4MIS.
**Keywords:** Semi-supervised learning, CNN, transformer, cross teaching.

## 1. Introduction

Medical image segmentation is a very basic and important step for computer-assisted diagnosis, treatment planning, and intervention (Wang et al., 2018; Luo et al., 2021d). Recently, Convolutional Neural Networks (CNNs) and Transformers with large-scale fine annotated images have achieved very promising results, even some applications have been used in the clinical flow (Shi et al., 2020; Chen et al., 2021c). But these methods almost require pixel/voxel-level expert labeling, which is more expensive and time-consuming than the natural image annotation (Yu et al., 2019). This dilemma makes semi-supervised segmentation a cheap and practical method to train powerful models with limited carefully labeled data and huge unlabeled or roughly labeled data. These proprieties can be used to accelerate clinical data annotation, model development, and even reduce the annotation that is often given by radiologists (Luo et al., 2021a,b; Xia et al., 2020; Wang et al., 2021a).
**Semi-supervised medical image segmentation:** Recently, semi-supervised learning has

raised high attention in the medical image computing community. A lot of semi-supervised methods have been proposed for medical image analysis, including pseudo-labelling (Wang et al., 2021a; Bai et al., 2017; Chen et al., 2021b), deep co-training (Qiao et al., 2018; Zhou et al., 2019), deep adversarial learning (Zhang et al., 2017; Hu et al., 2020), few-shot learning (Tang et al., 2021a), mean teacher and its extensions (Tarvainen and Valpola, 2017; Yu et al., 2019; Li et al., 2020; Reiß et al., 2021; You et al., 2021a,b), multi-task learning (Luo et al., 2021a; Kervadec et al., 2019; Chen et al., 2019), confidence learning (Vu et al., 2019), contrastive learning (Peng et al., 2021), and etc. All these methods combine both labeled and unlabeled data to train powerful and robust CNN models.

**CNNs *vs* Transformers for medical image segmentation:** CNN-based medical image segmentation approaches have been studied for many years, and most of them are based on UNet (Ronneberger et al., 2015) or its variants, achieving very promising results in various tasks (Isensee et al., 2021). Although the exceptional representation capacity, CNN-based methods are also limited by lacking the ability of modeling the global and long-range semantic information interaction, due to the intrinsic locality of convolution operations (Chen et al., 2021a). More recently, self-attention-based architectures (Dosovitskiy et al., 2020) (vision transformers) are introduced to the vision recognition tasks to model the long-range dependencies. After that, many variants of vision transformers achieved great success in natural image recognition tasks, like Swin-Transformer (Liu et al., 2021), DieT (Touvron et al., 2021), PVT (Wang et al., 2021b), TiT (Han et al., 2021), etc. Benefiting from the great representation capacity of transformers, several works attempt to use transformers to replace or combine CNNs for better medical image segmentation results, such as TransUNet (Chen et al., 2021a), Swin-UNet (Cao et al., 2021), CoTr (Xie et al., 2021), UNETR (Hatamizadeh et al., 2021), nnFormer (Zhou et al., 2021), etc. All these works show that transformers can further lead to performance gain than CNNs and also point out that it is worth to pay more attention to the transformer in the future. Although transformers have very exceptional representation capacity, it is still a data-hungry solution for recognition tasks, even require more data than CNNs (He et al., 2021; Tang et al., 2021b; You et al., 2022). How to train transformers with a semi-supervised fashion is also an interesting and challenging problem, especially for data limited medical image analysis tasks.

In this work, we present a simple yet efficient regularization scheme between CNN and Transformer, called Cross Teaching between CNN and Transformer. This framework takes both labeled and unlabeled images as inputs, and each input image passes a CNN and a transformer respectively to produce the prediction. For the labeled data, the CNN and transformer are supervised by the ground truth individually. Inspired by (Qiao et al., 2018; Han et al., 2018), we used predictions of unlabeled images generated by CNN/Transformer to update the parameters of the Transformer/CNN respectively. The advantages of the proposed are two-fold: (1) cross teaching is implicit consistency regularization, which can produce more stable and accurate pseudo labels than explicit consistency regularization. Explicit consistency regularization enforces to minimize the difference of different networks' predictions and optimize them at the same time, it could lead to predictions of different network are same but predictions are wrong. (2) this framework benefits from the two different learning paradigms, CNNs focus on the local information and transformers model the long-range relation, so the cross teaching can help to learn a unified segmenter with these two properties at the same time. The main **contributions** are two-fold: (1) We present a simple

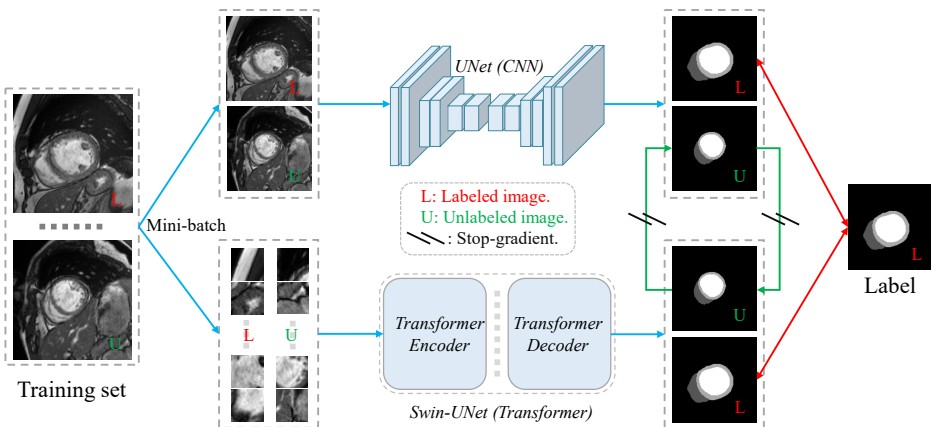

Figure 1: Overview of **Cross Teaching between CNN and Transformer**.

yet efficient cross teaching scheme for semi-supervised medical image segmentation. The proposed scheme implicitly encourages the consistency between different networks, when the advantages of CNNs and Transformers are leveraged to compensate each other for better performance; (2) To the best of our knowledge, this is the first attempt to use transformers to perform the semi-supervised medical image segmentation task and demonstrate it can outperform eight existing semi-supervised methods on a public benchmark.

## 2. Method

For the general semi-supervised learning, the training set always consists of two parts: labeled data set $D_N^l$ with $N$ annotated images and unlabeled data set $D_M^u$ with $M$ raw images ($M >> N$), the entire train set is $D_{N+M} = D_N^l \cup D_M^u$. For an image $x_i \in D^l$, its ground truth $y_i$ is available. In contrast, if $x_i \in D^u$, its ground truth is not provided. In this work, the proposed Cross Teaching between CNN and Transformer is depicted in Fig. 1. If $x_i \in D^l$, a commonly-used supervised loss function is used to update models' parameters. When $x_i$ belongs to $D^u$, we use a cross teaching strategy to cross supervise between a CNN ($f_\phi^c(.)$) and a Transformer ($f_\phi^t(.)$) for the updating of the parameters.

### 2.1. Cross teaching between CNN and Transformer

The original idea of cross teaching is inspired by three existing works: Deep Co-Training (Qiao et al., 2018), Co-Teaching (Han et al., 2018) and Cross Pseudo Supervision (Chen et al., 2021b). Deep Co-Training (Qiao et al., 2018) trains multiple deep neural networks with different views inputs and encourages view consistency for semi-supervised learning. Co-Teaching trains two deep neural networks simultaneously, and lets them teach each other in a mini-batch for noise-robust learning. Cross Pseudo Supervision (Chen et al., 2021b) trains two networks with the same architecture and different initializations to teach each other in a mini-batch for semi-supervised learning. All these methods introduce perturbations and encourage prediction to be consistent during the training stage. The differences are that Deep Co-Training uses input-level perturbation (multi-views), Co-Teaching utilizes

the supervision-level perturbation (noisy labels), and Cross Pseudo Supervision introduces perturbation in the network architecture-level. Here, we introduce the perturbation in both learning paradigm-level and output-level. For an input image $x_i$, the proposed framework produces two predictions:

$$p_i^c = f_\phi^c(x_i); \quad p_i^t = f_\phi^t(x_i) \tag{1}$$

where $p_i^c$, $p_i^t$ represent the prediction of a CNN ($f_\phi^c(.)$) and a Transformer ($f_\phi^t(.)$), respectively. As previously mentioned, CNN and Transformer are different learning paradigms for vision recognition, where CNN relies on the local convolution operation and the Transformer is based on the long-range self-attention, so these predictions have different properties essentially in the output level. Based on the predictions of $f_\phi^c(.)$ and $f_\phi^t(.)$, the pseudo labels for the cross teaching strategy are generated by this way:

$$pl_i^c = argmax(p_i^t) = argmax(f_\phi^t(x_i)); \quad pl_i^t = argmax(p_i^c) = argmax(f_\phi^c(x_i)) \tag{2}$$

where $pl_i^c$, $pl_i^t$ are generated pseudo labels for the CNN ($f_\phi^c(.)$) and the Transformer ($f_\phi^t(.)$) training, respectively. It's worthy to point that $pl_i^c$, $pl_i^t$ are pseudo segmentation results, and there is no gradient back-propagation between $p_i^c$ and $pl_i^c$, and between $p_i^t$ and $pl_i^t$ in each mini-batch. Then, the cross teaching loss for the unlabeled data is defined as:

$$\mathcal{L}_{ctl} = \underbrace{\mathcal{L}_{dice}(p_i^c, pl_i^c)}_{supervision\ for\ CNNs} + \underbrace{\mathcal{L}_{dice}(p_i^t, pl_i^t)}_{supervision\ for\ Transformers} \tag{3}$$

where $\mathcal{L}_{dice}$ is the standard dice loss function. Differently from consistency regularization loss, the cross teaching loss is a bidirectional loss function, one stream is from the CNN to the Transformer and the other is the Transformer to the CNN, there are no explicit constraints to enforce their predictions to become similar. In our framework, the transformer is also just used for complementary training, not used to produce final predictions.

## 2.2. The overall objective function

The overall training objective function is a joint loss with two parts, a supervised loss on the labeled data and an unsupervised loss for the unlabeled data. The supervised loss $\mathcal{L}_{sup}$ consists of two widely-used loss functions:

$$\mathcal{L}_{sup} = \mathcal{L}_{ce}(p_i, y_i) + \mathcal{L}_{dice}(p_i, y_i) \tag{4}$$

where $\mathcal{L}_{ce}$, $\mathcal{L}_{dice}$ are the cross-entropy loss and dice loss, respectively. $p_i$, $y_i$ represent the prediction and label of image $x_i$. The overall objective is defined as :

$$\mathcal{L}_{total} = \mathcal{L}_{sup} + \lambda\mathcal{L}_{ctl} \tag{5}$$

where $\lambda$ is a weight factor, which is defined by a time-dependent Gaussian warming up function commonly (Yu et al., 2019; Luo et al., 2021a; Luo, 2020): $\lambda(t) = 0.1 \cdot e^{(-5(1-\frac{t_i}{t_{total}})^2)}$, where $t_i$ denotes the current training iteration and $t_{total}$ is the total iteration number.

## 3. Experiments

### 3.1. Dataset and evaluation metrics

In this work, all experiments and comparisons are based on the public benchmark dataset ACDC (Bernard et al., 2018). The ACDC dataset contains 200 annotated short-axis cardiac cine-MR images from 100 patients. The segmentation masks of the left ventricle (LV), myocardium (Myo), and right ventricle (RV) are provided for clinical and algorithm research. 140 images from 70 patients, 60 images from 30 patients are randomly selected for training and validation respectively. Due to the large inter-slice spacing, 2D segmentation is more suitable than direct 3D segmentation (Bai et al., 2017). For the pre-processing, we resize all the slices into 256×256 pixels and re-scale the intensity of each slice to [0, 1]. We used standard data augmentation to enlarge training sets' scale and avoid over-fitting, including random cropping with a 224×224 patch, random rotating between -25 and 25 degrees, random flipping. At the inference stage, predictions are generated slice by slice and stacked into a 3D volume. For a fair comparison, we don't use any post-processing strategy. Two commonly-used metrics are employed to quantitatively evaluate the 3D segmentation results: 1) Dice Coefficient (DSC); 2) 95% Hausdorff Distance ($HD_{95}$). (Details in Sec.A.1).

### 3.2. Implementation details

**Network architectures and training details:** There are two types of networks in the proposed method (see Fig. 1): a CNN-based segmentation network UNet (Ronneberger et al., 2015) and a Transformer segmentation Swin-UNet (Cao et al., 2021). Both of them are U-shape-based networks, but they are based on two different learning paradigms. For a fair comparison, in this work, we employ the open-sourced implementation of UNet (Ronneberger et al., 2015) and Swin-UNet (Cao et al., 2021) as baselines. We use PyTorch (Paszke et al., 2019) for all method's implementations, and run all experiments on a Ubuntu desktop with a GTX1080TI GPU. All these networks are trained by the SGD optimizer with a batch size of 16, where half of them are labeled in batch for semi-supervised learning. The poly learning rate strategy is used to adjust the learning rate, where the initial learning rate is set to 0.01. All implementation are available at: https://github.com/HiLab-git/SSL4MIS.
**Ablation study:** In this work, we first investigate the performance when using the Transformers for semi-supervised learning directly. We use the Swin-UNet (Cao et al., 2021) as backbones to replace the UNet (Ronneberger et al., 2015) for semi-supervised learning in Mean Teacher (MT) (Tarvainen and Valpola, 2017), Entropy Minimization (Ent Min) (Vu et al., 2019), Deep Adversarial Network (DAN) (Zhang et al., 2017), Deep Co-Training (DCT) (Qiao et al., 2018). Then, we further investigate the results when cross teaching between different architectures, including CNN and Transformer, CNNs and CNN and Transformer and Transformers. Finally, we investigate the performance gain of different cross teaching loss functions, including cross-entropy loss, dice loss, and compare them against classical consistency regularization (Tarvainen and Valpola, 2017; Yu et al., 2019).
**Comparison with baselines and existing methods:** To demonstrate the effectiveness of the proposed method, we compared our method against baselines and eight recently semi-supervised methods. Firstly, we investigate the upper-bound/low-bound performances of UNet and Swin-UNet based on all/limited labeled images, respectively, referred to as

Table 1: Ablation study results when using 7 cases as labeled. RV, Myo, LV represent the right ventricle, myocardium and left ventricle, respectively. The first section shows the results when using transformers for semi-supervised segmentation directly. The second section lists the results when using different architectures and supervision strategies, where CT and CR mean the cross teaching and consistency regularization, respectively. The last section shows the effects when using different cross-teaching loss functions. $^*$ means the predictions are produced by Swin-UNet, the others are based on UNet. Blue numbers mean the best results.

| Method | RV DSC | RV $HD_{95}$ | Myo DSC | Myo $HD_{95}$ | LV DSC | LV $HD_{95}$ | Mean DSC | Mean $HD_{95}$ |
|---|---|---|---|---|---|---|---|---|
| LS$^*$ | 0.42(0.21) | 34.6(24.6) | 0.499(0.18) | 19.0(12.2) | 0.617(0.23) | 23.6(13.4) | 0.512(0.207) | 25.7(16.8) |
| MT$^*$ | 0.433(0.231) | 25.3(19.6) | 0.486(0.179) | 18.2(11.9) | 0.614(0.234) | 22.0(13.0) | 0.511(0.215) | 21.8(14.8) |
| DAN$^*$ | 0.504(0.208) | 24.0(18.0) | 0.438(0.159) | 18.5(14.2) | 0.643(0.2) | 32.3(27.6) | 0.528(0.189) | 24.9(19.9) |
| DCT$^*$ | 0.465(0.223) | 29.0(22.2) | 0.499(0.175) | 17.5(12.1) | 0.622(0.22) | 22.5(13.0) | 0.528(0.206) | 23.0(15.8) |
| EM$^*$ | 0.456(0.216) | 32.1(24.2) | 0.51(0.176) | 18.9(11.7) | 0.62(0.223) | 23.9(12.3) | 0.529(0.205) | 25.0(16.1) |
| FS$^*$ | 0.785(0.137) | 11.4(13.7) | 0.779(0.083) | 5.6(7.4) | 0.863(0.123) | 7.4(9.8) | 0.809(0.115) | 8.1(10.3) |
| CNN&CNN(CT) | 0.791(0.189) | 13.2(15.7) | 0.821(0.067) | 8.4(11.6) | 0.886(0.092) | 11.5(16.2) | 0.833(0.116) | 11.0(14.5) |
| Trans&Trans(CT)$^*$ | 0.805( 0.117) | 12.9(18.0) | 0.779(0.072) | 6.1(7.2) | 0.856(0.119) | 12.2(15.3) | 0.813(0.103) | 10.4(13.5) |
| CNN&Trans(CR) | 0.782(0.195) | 14.4(15.3) | 0.806(0.082) | 12.7(15.5) | 0.87(0.111) | 18.1(21.3) | 0.82(0.129) | 15.1(17.4) |
| CNN&Trans(**proposed**) | 0.848(0.112) | 7.80(7.6) | 0.844(0.052) | 6.9(9.2) | 0.901(0.085) | 11.2(14.8) | 0.864(0.083) | 8.6(10.5) |
| Ours(CE) | 0.807(0.183) | 9.8(11.0) | 0.829(0.066) | 7.1(9.1) | 0.881(0.106) | 14.4(17.8) | 0.839(0.118) | 10.5(12.6) |
| Ours(CE)$^*$ | 0.829(0.103) | 10.6(16.0) | 0.802(0.065) | 5.1(5.6) | 0.875(0.109) | 11.5(17.4) | 0.835(0.092) | 9.1(13.0) |
| Ours(DICE) | 0.848(0.112) | 7.80(7.6) | 0.844(0.052) | 6.9(9.2) | 0.901(0.085) | 11.2(14.8) | 0.864(0.083) | 8.6(10.5) |
| Ours(DICE)$^*$ | 0.85(0.099) | 9.2(15.2) | 0.825(0.056) | 5.4(7.2) | 0.893(0.09) | 8.4(12.9) | 0.856(0.082) | 7.7(11.8) |
| Ours(CE+DICE) | 0.84(0.142) | 8.8(9.6) | 0.849(0.045) | 6.0(7.7) | 0.9(0.089) | 12.4(18.9) | 0.863(0.092) | 9.1(12.1) |
| Ours(CE+DICE)$^*$ | 0.844(0.098) | 8.1(11.6) | 0.815(0.06) | 5.6(7.8) | 0.886(0.097) | 9.1(13.1) | 0.848(0.085) | 7.6(10.8) |

full/limited supervisions (FS/LS). Then, we compared with eight previous semi-supervised methods: 1) Mean Teacher (MT) (Tarvainen and Valpola, 2017), 2) Entropy Minimization (EM) (Vu et al., 2019), 3) Deep Adversarial Network (DAN) (Zhang et al., 2017), 4) Uncertainty Aware Mean Teacher (UAMT) (Yu et al., 2019), 5) Interpolation Consistency Training (ICT) (Verma et al., 2019), 6) Cross Pseudo Supervision (CPS) (Chen et al., 2021b), 7) Cross Consistency Training (CCT) (Ouali et al., 2020), 8) Deep Co-Training (DCT) (Qiao et al., 2018). All these methods used the same backbone and were trained and tested with the same settings, notably all these methods are open available (Luo, 2020).

## 4. Results

**Ablation study:** Tab. 1 shows the results of ablation study. The first section presents the results of using Swin-UNet as segmentation networks to perform semi-supervised segmentation rather than UNet. It can be found that compared with UNet, Swin-UNet performs quite badly when combing previous semi-supervised approaches (results of UNet are presented in Tab. 2). The reason may be transformers are data-hungry approaches, directly transferring them to perform semi-supervised learning may be not suitable. The second section lists the comparison results when using cross teaching to supervise different combinations. It can be observed that using cross teaching between CNN and Transformer can achieve better results than others combinations. It demonstrates that CNN and Transformer with different learning paradigm can compensate each other in the training stage. In addition, the results also demonstrated that cross teaching strategy outperforms the consistency regularization. Finally, we investigate the impact of different cross-teaching loss functions. The last section shows that dice loss can further improve the performance than cross-entropy loss, but the joint loss of cross-entropy and dice can not lead to more gain. In addition, we also investigated the performance of the Transformer branch and the ensemble of the Transformer

Table 2: Mean 3D $DSC$ and $HD_{95}$ (mm) on the ACDC dataset. All results are based on the same backbone (UNet) with a fixed seed. Mean and standard variance (in parentheses) are presented in this table. Red numbers denote the p-value $< 0.05$ based on paired t-test when comparing with existing methods. The performance of nnUNet is borrowed from ACDC leaderboard.

| Labeled | Method | RV | | Myo | | LV | | Mean | |
|---|---|---|---|---|---|---|---|---|---|
| | | $DSC$ | $HD_{95}$ | $DSC$ | $HD_{95}$ | $DSC$ | $HD_{95}$ | $DSC$ | $HD_{95}$ |
| 3 cases | LS | 0.37(0.32) | 44.4(28.4) | 0.548(0.287) | 24.4(19.9) | 0.618(0.329) | 24.3(21.4) | 0.512(0.312) | 31.0(23.2) |
| | MT | 0.403(0.291) | 53.9(28.7) | 0.586(0.251) | 23.1(20.5) | 0.709(0.246) | 26.3(25.8) | 0.566(0.263) | 34.5(25.0) |
| | DAN | 0.378(0.318) | 39.6(25.7) | 0.568(0.259) | 25.8(20.3) | 0.64(0.306) | 32.4(27.9) | 0.528(0.294) | 32.6(24.6) |
| | DCT | 0.413(0.305) | 31.7(20.2) | 0.617(0.235) | 20.3(18.0) | 0.717(0.254) | 27.3(24.5) | 0.582(0.265) | 26.4(20.9) |
| | EM | 0.447(0.312) | 32.4(22.7) | 0.628(0.235) | 19.0(20.3) | 0.731(0.244) | 20.9(20.2) | 0.602(0.264) | 24.1(21.0) |
| | UAMT | 0.508(0.328) | 35.4(24.8) | 0.615(0.26) | 19.3(21.8) | 0.707(0.273) | 22.6(19.8) | 0.610(0.287) | 25.8(22.1) |
| | ICT | 0.448(0.327) | 23.8(16.4) | 0.620(0.24) | 20.4(20.4) | 0.673(0.286) | 24.1(21.1) | 0.581(0.284) | 22.8(19.3) |
| | CCT | 0.408(0.322) | 34.2(23.6) | 0.647(0.206) | 22.4(19.4) | 0.704(0.239) | 27.1(23.9) | 0.586(0.256) | 27.9(22.3) |
| | CPS | 0.438(0.306) | 35.8(24.1) | 0.652(0.213) | 18.3(16.3) | 0.72(0.248) | 22.2(22.0) | 0.603(0.256) | 25.5(20.8) |
| | **Ours** | 0.577(0.338) | 21.4(20.2) | 0.628(0.251) | 11.5(11.7) | 0.763(0.244) | 15.7(15.5) | 0.656(0.278) | 16.2(15.8) |
| 7 cases | LS | 0.649(0.31) | 19.4(19.5) | 0.787(0.087) | 12.2(13.8) | 0.856(0.117) | 17.4(18.6) | 0.764(0.171) | 16.3(17.3) |
| | MT | 0.769(0.198) | 17.9(21.3) | 0.794(0.093) | 10.8(13.8) | 0.866(0.105) | 14.5(17.6) | 0.810(0.132) | 14.4(17.6) |
| | DAN | 0.763(0.212) | 13.5(15.1) | 0.784(0.105) | 9.8(10.0) | 0.840(0.138) | 20.6(21.0) | 0.795(0.152) | 14.6(15.4) |
| | DCT | 0.750(0.221) | 15.3(15.6) | 0.793(0.098) | 10.7(12.6) | 0.870(0.099) | 15.5(19.2) | 0.804(0.14) | 13.8(15.8) |
| | EM | 0.729(0.238) | 15.3(14.7) | 0.79(0.099) | 12.4(16.2) | 0.855(0.122) | 15.8(17.4) | 0.791(0.153) | 14.5(16.1) |
| | UAMT | 0.775(0.202) | 11.5(10.9) | 0.801(0.092) | 13.7(24.9) | 0.871(0.103) | 18.1(20.9) | 0.815(0.132) | 14.4(18.9) |
| | ICT | 0.755(0.232) | 11.4(11.5) | 0.807(0.082) | 8.7(9.2) | 0.871(0.101) | 14.0(17.3) | 0.811(0.138) | 11.4(12.7) |
| | CCT | 0.766(0.198) | 14.3(16.0) | 0.812(0.071) | 10.4(13.4) | 0.87(0.101) | 14.6(18.6) | 0.816(0.127) | 13.1(16.0) |
| | CPS | 0.791(0.189) | 13.2(15.7) | 0.821(0.067) | 8.4(11.6) | 0.886(0.092) | 11.5(16.2) | 0.833(0.116) | 11.0(14.5) |
| | **Ours** | 0.848(0.112) | 7.80(7.6) | 0.844(0.052) | 6.9(9.2) | 0.901(0.085) | 11.2(14.8) | 0.864(0.083) | 8.60(10.5) |
| Total | FS | 0.900(0.075) | 4.40(4.9) | 0.893(0.028) | 2.4(4.0) | 0.941(0.048) | 4.0(9.7) | 0.911(0.05) | 3.60(6.2) |
| | nnUNet | 0.925 | 10.1 | 0.908 | 7.4 | 0.948 | 6.2 | 0.927 | 7.9 |

and CNN branches. We found that the Transformer branch achieves very similar results compared with the CNN branch, and the ensemble result outperforms both the Transformer and CNN branches. Still, the transformer and ensemble results require more computation cost, as the transformer has more parameters ($27.12M$ $vs$ $1.81M$). So, we use the CNN branch outputs to compare with existing methods fairly. (Details in Sec. A.2, B.1 and B.2)

**Comparison with baselines and existing works:** Tab. 2 lists the results of all methods on the ACDC dataset when using 3 cases (6 volumes) and 7 cases (14 volumes) as labeled samples. Note that, all differences between these methods just exist in the training stage. In the inference stage, these methods just used the trained UNet to produce final predictions and ignored all auxiliary training modules, and also did not use any ensemble strategies. It can be found that the proposed cross-training between CNN and Transformer achieves better results than the other 8 methods in a large margin when used 3 and 7 labeled cases. In the setting of 7 labeled cases, the proposed achieved a significant gain than the second method (cross pseudo supervision strategy), in terms of $3.8\%$ in $DSC$ and $3.6mm$ in $HD_{95}$ respectively. Furthermore, despite the labeled sample being very limited (3 cases are labeled, which is less than $10\%$ of the training set), our proposed method remains is superior to other methods, with more than $4\%$ of mean $DSC$ improvement and $6mm$ of mean $HD_{95}$ decrease. These results show the potential of the proposed method which can alleviate the label cost by learning from the limited data and large-scale unlabeled data. In addition, compared with existing semi-supervised learning methods, the proposed framework consists of very simple training strategies and common components with very low complexity, it is more desirable to apply it in clinical practice. Although the proposed outperforms existing semi-supervised methods, it also can't achieve a comparable results compared with the state-of-the-art (SOTA) fully-supervised method (Isensee et al., 2021). It shows that using semi-supervised methods to achieve SOTA remains an important yet challenging problem. Fig. 2 shows some visualization comparisons between various methods when using 3 labeled

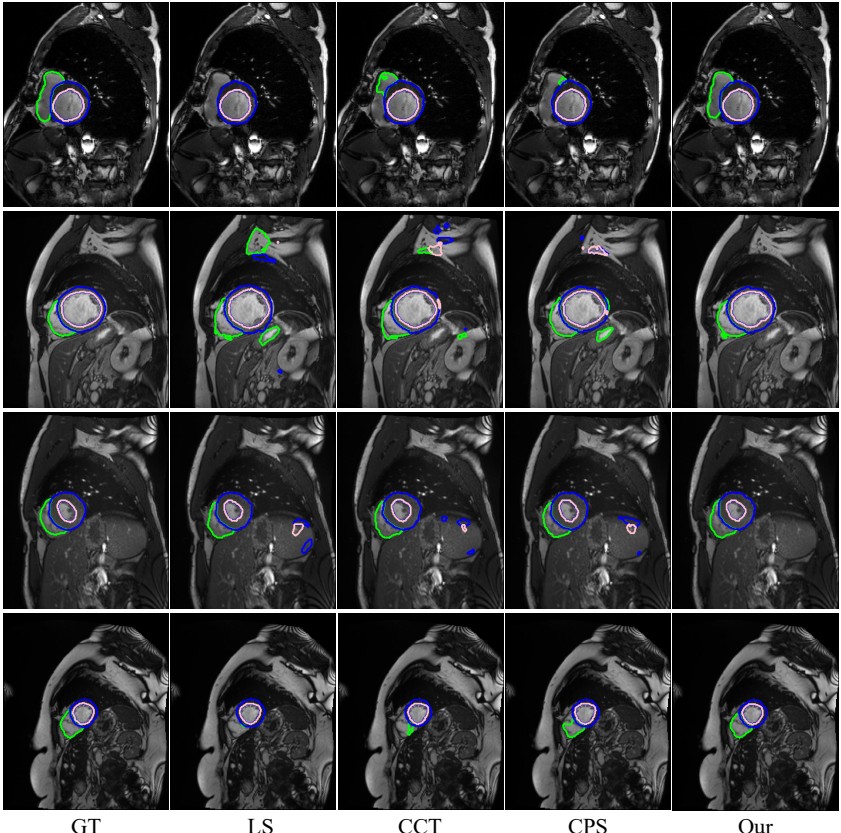

GT        LS        CCT        CPS        Our

Figure 2: Visualization comparison of different methods on validation images. The first two rows used 3 labeled cases and the last two rows used 7 labeled cases.

cases and 7 labeled cases. Compared with CCT (Ouali et al., 2020) and CPS (Chen et al., 2021b), our method can produce more plausible segmentation with fewer false-positive and missing-segmentation regions.

## 5. Conclusion

In conclusion, this work introduces the Transformer to the semi-supervised medical image segmentation. To achieve this goal, inspired by co-teaching (Han et al., 2018) and cross pseudo supervision (Chen et al., 2021b), we present a cross teaching between CNN and Transformer to utilize the unlabeled data. The idea is based on the assumption that CNN can capture local features efficiently and Transformer can model the long-range relation better, and these properties can complement each other during training. Experimental results on an open-benchmark showed that the proposed can outperform eight existing semi-supervised learning methods. In the future, we will combine some advanced techniques (Tang et al., 2021b) with this method to further reduce the annotation costs, especially for the dense-annotation-based multi-organ segmentation tasks (Luo et al., 2021c).

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

## Appendix A. Details of experiments

### A.1. Dataset

The ACDC dataset[1] consists of 200 short-axis cine-MRI scans from 100 patients, and each patient has two annotated scans corresponding to end-diastolic (ED) and end-systolic (ES) phases. In this paper, we split the dataset based on patient-id into the training set (70 patients) and validation set (30 patients), so the training set has 140 scans (70 ED scans and 70 ES scans), and the validation set consists of 60 scans (30 ED scans and 30 ES scans). For semi-supervised learning, we randomly select 3 patients (6 scans) and 7 patients (14 scans) as labeled data ($\approx 5\%$ and $10\%$ of the training set) and the remaining 67 patients (144 scans) and 63 patients (126 scans) are seen as unlabeled data.

### A.2. Training details

We used UNet (Ronneberger et al., 2015) and Swin-UNet (Cao et al., 2021) as segmentation networks respectively, their PyTorch implementations borrowed from widely-used projects PyMIC[2] and Swin-UNet[3]. All experiments of this paper were run on a Ubuntu 16.04 desktop with a GTX1080TI GPU and the PyTorch1.8.1 library. All models are optimized using stochastic gradient descent (SGD) with the polylearning rate strategy, where the initial learning rate was set to 0.01. The total iterations are 30k, and we used the latest checkpoint for testing and reporting results (Yu et al., 2019). For a fair comparison, all existing semi-supervised segmentation methods used UNet (Ronneberger et al., 2015) as the segmentation network and also used the same training strategies (data augmentation methods, total iterations, and other training hyper-parameters). To reproduce these results, all methods' training and testing code and processed data are available at: https://github.com/HiLab-git/SSL4MIS.

## Appendix B. Results analysis

### B.1. Computational-cost

To compare the computational cost among the proposed and others comparisons, we investigated the number of forwarding pass times of an input image in one iteration (*FTimes*), the total training time (*TTimes*), per case inference time (*ITimes*), when using same software and hardware settings. Table 3 lists the quantitative comparison of computational-cost. Note that all existing methods used UNet as a segmentation network, and our method used UNet and Swin-UNet at the same time, so we listed UNet/Swin-UNet/ensemble of UNet and Swin-UNet inference time, respectively. It can be found that our method requires more training time than others, as the Swin-UNet consists of more parameters. But all methods require very similar inference costs when using the same segmentation network. For our method, the transformer branch needs more inference time than the CNN branch due to the transformer with more parameters. The ensemble results spend the most inference time than other methods, as it runs two models to generate predictions simultaneously.

---

1. https://www.creatis.insa-lyon.fr/Challenge/acdc/index.html

2. https://github.com/HiLab-git/PyMIC

3. https://github.com/HuCaoFighting/Swin-Unet

Table 3: Comparison of computational cost between our method and existing methods based on 7 labeled cases. *FTimes* means the times of an input image passed the networks during one iteration. *TTime* means the total training time (hours). *ITime* means per case inference time (s). For our method, we listed the inference time of CNN (pink number), Transformer (lime number), and their ensemble (yellow number), respectively.

|  | LS | MT | DAN | DCT | EM | UAMT | ICT | CCT | CPS | Ours |
|---|---|---|---|---|---|---|---|---|---|---|
| *FTimes* | 1 | 2 | 2 | 2 | 1 | 9 | 3 | 1 | 2 | 2 |
| *TTime(h)* | 4.17 | 4.37 | 5.33 | 4.83 | 4.50 | 5.61 | 5.15 | 5.08 | 5.17 | 6.22 |
| *ITime(s)* | 0.46 | 0.46 | 0.46 | 0.46 | 0.46 | 0.46 | 0.46 | 0.46 | 0.46 | 0.46/0.57/0.78 |

Table 4: Mean 3D *DSC* and $HD_{95}$ (mm) on the ACDC dataset. These results are based on our proposed method, Ours (CNN), Ours (Trans), and Ours CNN & Trans) represent the result of the CNN branch and the Transformer branch and their ensemble, respectively. Mean and standard variance (in parentheses) are presented in this table.

| Labeled | Method | RV | | Myo | | LV | | Mean | |
|---|---|---|---|---|---|---|---|---|---|
| | | $DSC$ | $HD_{95}$ | $DSC$ | $HD_{95}$ | $DSC$ | $HD_{95}$ | $DSC$ | $HD_{95}$ |
| 3 cases | Ours(CNN) | 0.577(0.338) | 21.4(20.2) | 0.628(0.251) | 11.5(11.7) | 0.763(0.244) | 15.7(15.5) | 0.656(0.278) | 16.2(15.8) |
| | Ours(Trans) | 0.660(0.198) | 23.1(20.7) | 0.597(0.201) | 10.9(9.6) | 0.748(0.207) | 12.4(10.8) | 0.669(0.202) | 15.5(13.7) |
| | Ours(Trans & CNN) | 0.648(0.283) | 16.5(12.7) | 0.638(0.244) | 10.5(11.1) | 0.777(0.236) | 11.6(13.5) | 0.688(0.254) | 12.8(12.4) |
| 7 cases | Ours(CNN) | 0.848(0.112) | 7.80(7.6) | 0.844(0.052) | 6.9(9.2) | 0.901(0.085) | 11.2(14.8) | 0.864(0.083) | 8.60(10.5) |
| | Ours(Trans) | 0.850(0.099) | 9.20(15.2) | 0.825(0.056) | 5.4(7.2) | 0.893(0.09) | 8.40(12.9) | 0.856(0.082) | 7.70(11.8) |
| | Ours(Trans & CNN) | 0.865(0.094) | 6.4(6.0) | 0.851(0.049) | 4.7(6.2) | 0.909(0.078) | 7.5(12.8) | 0.875(0.074) | 6.2(8.3) |

## B.2. Performance of CNN branch and Transformer branch

We also investigated the difference of performance in the CNN branch and Transformer branch in 3 and 7 labeled cases settings. The results are presented in Table 4. It can be found that the CNN branch and Transformer branch achieve very similar performance, but as Sec.B.1 shows, the Transformer branch requires more inference cost. We further investigated the performance when ensembling the prediction of CNN and Transformer, the ensemble prediction is defined as $argmax(\frac{f_\phi^t(x_i)+f_\phi^c(x_i)}{2})$. The result was listed in the lost row of Table 4

