# OpenReview forum: "Semi-Supervised Medical Image Segmentation via Cross Teaching between CNN and Transformer"
_MIDL.io/2022/Conference — MIDL 2022_

### Official Review · Reviewer_xdsd · 2022-01-21

**Confidence:** 5
**Preliminary Rating:** 3
**Recommendation:** Poster

**Summary:**

This paper proposed a cross-teaching scheme between CNN and Transformer for medical image segmentation. In addition to classical supervised learning, the prediction of CNN will be used for regulating Transformer, and the prediction of Transformer used for regulating CNN for unlabeled data. As a whole framework, it belongs to semi-supervised learning methods. The experiments were conducted on the public benchmark dataset ACDC.

**Strengths:**

+ The method is easy to follow and implement. The underlying motivation of the cross-teaching is clear.
+ The baseline choice and comparison are rich.
+ The statistical significance of the results is reported.
+ Both quantitative and qualitative results are presented.
+ The public codebase of this work seems strong and comprehensive.

**Weaknesses:**

- The results were only validated on one dataset. The generalizability of the framework to other segmentation tasks remains unclear.
- The ablation study is incomplete. Several variables in comparison were not controlled.
- The experimental settings were not clearly stated.
- The computational cost and training speed of the proposed method vs. existing methods were not mentioned in the paper.

**Deanonymize Review:**

no

**Detailed Comments:**

Table 1 and Table 2 look crowded; I suggest making standard deviations smaller font sizes.

**Final Rating After The Rebuttal:**

4: Weak Accept

**Justification Of The Final Rating:**

In general, I think it is a nice work that co-trains CNN and Transformer for cardiac structure segmentation, although the generalizability of this training paradigm is unclear due to the lack of applications. More investigation is needed. The public released code seems well-organized, so I'd like to recommend a weak accept to this work.

See detailed comments in my following-up responses.

**Paper Type:**

methodological development

**Questions To Address In The Rebuttal:**

Overall, I like the idea of cross-teaching between CNN and Transformer, and the results seem promising on the ACDC dataset. However, I must hold the significance of results and generalizability of conclusions before seeing the answers to the following questions:

1. In the second section of Table 1, what does “ours” stand for? What I understood was “ours” is equivalent to “CT” (cross-teaching) as indicated in Sec. 2.1.

2. In the second section of Table 1, why does only the 2nd row was produced by Swin-UNet? Is the comparison fair and controlled?

3. Please include the benchmark performance (on public websites) in Table 2 as a reference. Since every performance in Table 2 was obtained by the authors (correct me if I am wrong), it is sensible to ensure that the reported performance is comparable or superior to the current state of the arts reported by other groups.

4. The authors claimed in the abstract that the proposed method is simpler than existing ones, but at least in terms of model parameters, it was much larger than other methods (because of CNN and Transformer). Please report the training time, the number of parameters in the revised version.

5. Why only CNN was used for testing? Why not Transformer-only or an ensemble of CNN&Transformer? The authors mentioned concerns about computational cost, but performance is often more important than computational cost. Only with comparable performance, we prefer the computational-friendly solution. Therefore, I would like to see a direct comparison among CNN-only, Transformer-only, and CNN&Transformer in the test phase.


**Special Issue:**

no

---

### Official Review · Reviewer_tmso · 2022-01-24

**Confidence:** 3
**Preliminary Rating:** 2
**Recommendation:** Poster

**Summary:**

This paper concerns the problem of limited training in CNNs and transformers in medical image segmentation.  It proposes a semi-supervised method for cross-teaching between a CNN and a transformer.  It uses labeled and unlabeled data and trains both CNN (UNet) and transformer (Swin-UNet) to take advantage of their local and long-range perspectives, and shares information between them.  It outperforms eight existing semi-supervised methods on a public benchmark.

**Strengths:**

This is a very simple approach, easy to understand
It addresses the problem of limited training, which is a ubiquitous problem in medical image analysis.
It uses the positive aspects of both CNN and transformer to good effect


**Weaknesses:**

The exclusive use of ACDC for such a simple and generally applicable approach leaves one wondering about other tasks. While impressive in its ability to beat the other methods, its potential widespread use remains in question. None of the results compare favorably to the current ACDC leaderboard (and this comparison is not made by the authors).
There have been many improvements to the basic UNet architecture over the years.
The method is compared only to other semi-supervised methods.


**Deanonymize Review:**

no

**Detailed Comments:**

Proprieties should be properties
Low-bound should be lower-bound
The details describing the semi-supervision aspect of the experiments are sketchy.  In particular, it is noted that 140 images from 70 patients are used for training and 60 images from 30 patients are used for validation.  By “validation” do the authors mean “testing”? Is cross-validation used (it is not mentioned)?   How much fully labeled data are used versus no labels?  Where is the semi-supervised aspect?
Why resize the images?  What happens to the labels as the result of resizing (as there must be interpolation, which is problematic with labels)?
“when combing previous” should be “when comparing to previous”
“this work attempts to introduce” should be “this work introduces”


**Paper Type:**

methodological development

**Questions To Address In The Rebuttal:**

How does the method perform on other segmentation problems?
Compare approach using full supervision versus semi-supervision (with a sequence of experiments)
Why does the method not compare well to the current ACDC leading methods?


**Special Issue:**

no

---

### Official Review · Reviewer_UgPH · 2022-01-24

**Confidence:** 3
**Preliminary Rating:** 3
**Recommendation:** Poster

**Summary:**

A simple but efficient cross-teaching framework using the existing CNN model and transformer is proposed to tackle the semi-supervised medical image segmentation. Though it is an incremental work by introducing the hybrid transformer and convolution structure, the performance achieves the SOTA compared with other basslines.

**Strengths:**

The extensive experiments show the performance superiority of hybrid networks in medical segmentation compared with other baselines. Ablation studies are also well-analyzed. The code will be publicly released for reproduction. The paper is well-organized and easy to follow.

**Weaknesses:**

It is better to discuss more recent works.
[1]. Recurrent Mask Refinement for Few-Shot Medical Image Segmentation
[2]. Coarse-to-fine adversarial networks and zone-based uncertainty analysis for NK/T-cell lymphoma segmentation in CT/PET images.
[3]. Every Annotation Counts: Multi-Label Deep Supervision for Medical Image Segmentation.

The novelty is limited by introducing the existing transformer (swin transformer) and U-net structure. As the main novelty, the cross-teaching strategy is also simple and has not changed a lot compared with other semi-supervised learning.

The transformer has a powerful ability to extract global and long-range knowledge. But it suffers from memory and computational cost. It is better to provide the time and GPU memory of the network both in the training and test stage.

The structure of the proposed network is 2D network which is blind to the spatial relationships among the slices. It is better to verify the superiority of the proposed network than other 3D models.


**Deanonymize Review:**

no

**Final Rating After The Rebuttal:**

4: Weak Accept

**Justification Of The Final Rating:**

After reviewing all rebuttal material and discussion, I am happy to change my rate to weak accept.  Thanks for the authors' clarification to dismiss some questions. I hope this work could inspire some research in this community.

**Paper Type:**

both

**Questions To Address In The Rebuttal:**

Clarity and emphasize the main novelty of the paper.

To better verify the superiority of the proposed network, it is better to compare it with other SOTA 3D models.

Analyze the inference time and computation cost both in the training and test stage.




**Special Issue:**

no

---

### Official Review · Reviewer_5ere · 2022-01-24

**Confidence:** 4
**Preliminary Rating:** 4
**Recommendation:** Oral, Poster

**Summary:**

This paper describes an approach to semi-supervised segmentation of images using cross teaching between a standard convolutional neural network (CNN) and a transformer network. These two approaches are potentially complementary since CNNs are composed of localized convolutions, while transformer networks compute embeddings derived from self-attention that can involve long-range interactions. The two networks are combined in a relatively straightforward cross teaching approach and the approach is evaluated on a public data set of  cardiac MRI (ACDC). The results are compared to a number of baseline semi-supervised approaches, as well as ablative variations of the proposed technique.  This is a nice paper overall that makes a novel contribution to the segmentation literature, but is likely not that impactful to the specific application of cardiac imaging.

**Strengths:**

- The idea of using a transformer and a CNN in cross teaching semi-supervised paradigm is clever.
- Multiple techniques are used for comparison.
- The results seem to show good performance compared to baseline techniques.

**Weaknesses:**

- The application to cardiac MRI is a bit of an afterthought. There is nothing in the technique that is tailored to the application, and the details of how the data set is used are lacking.
- There are multiple aspects of the experiments that are not well explained.

**Deanonymize Review:**

no

**Detailed Comments:**

- The ACDC data set is actually a 4D data set. I believe the labels are provided at end systole and end diastole. It is not clear which labels the paper are using and whether the rest of the cine frames that do not have labels are being used or ignored.
- The experiments describe how many labeled results are used but I could not find how many unlabeled images were used for semi-supervised training. Is it 2 * (70 - the number of cases)?  Please clarify.
- In Table 1, are the fully supervised results using 7 cases since that is stated in the caption? I believe limited supervision is using 7 cases and fully supervised is using 70 but that should be made explicit.
- It is not clear what the differences are in the results in Tables 1 and the 7 case results of Table 2.
- Why are the transformer outputs ignored at test-time? How does it perform compared to the CNN? Is that the starred results in the table? It is confusing because the starred results refer to many different methods.
- References are difficult to review since the bibliography format lists the first name of the author at the beginning, but the sorting is based on the last name of the author.



**Final Rating After The Rebuttal:**

4: Weak Accept

**Justification Of The Final Rating:**

Authors clarified some of the questions related to experimental design and results. My impression of the paper remains mostly positive and my rating remains unchanged. This paper proposes an interesting, but not groundbreaking, technique and the growing interest in transformer models will likely make this work attractive to the MIDL audience.

**Paper Type:**

methodological development

**Questions To Address In The Rebuttal:**

- The ACDC data set is actually a 4D data set. I believe the labels are provided at end systole and end diastole. It is not clear which labels the paper are using and whether the rest of the cine frames that do not have labels are being used or ignored.
- The experiments describe how many labeled results are used but I could not find how many unlabeled images were used for semi-supervised training. Is it 2 * (70 - the number of cases)?  Please clarify.
- In Table 1, are the fully supervised results using 7 cases since that is stated in the caption? I believe limited supervision is using 7 cases and fully supervised is using 70 but that should be made explicit.
- It is not clear what the differences are in the results in Tables 1 and the 7 case results of Table 2.
- Why are the transformer outputs ignored at test-time? How does it perform compared to the CNN? Is that the starred results in the table? It is confusing because the starred results refer to many different methods.


**Special Issue:**

no

---

### Meta-Review · Area_Chair_ovGj · 2022-02-17

**Recommendation:** Accept (Poster)
**Confidence:** 4

**Metareview:**

This paper proposed a simple framework by combining CNN and Transformer by cross teaching for semi-supervised medical image segmentation. Experiments on cardiac application demonstrate better performance over other methods. Overall, this paper is well organized and the results are convincing. The majority of reviewers (3/4) has recommended weak accept. Based on own reading, I also recommend accept, although comparison results with other SOTA methods such as 3D models can be added in the final version.

---

### Decision · Program_Chairs · 2022-02-28

Accept